# PAUF Induces Migration of Human Pancreatic Cancer Cells Exclusively via the TLR4/MyD88/NF-κB Signaling Pathway

**DOI:** 10.3390/ijms231911414

**Published:** 2022-09-27

**Authors:** So Eun Youn, Fen Jiang, Hye Yun Won, Da Eun Hong, Tae Heung Kang, Yun-Yong Park, Sang Seok Koh

**Affiliations:** 1Department of Biomedical Sciences, Dong-A University, Busan 49315, Korea; 2Innovative Discovery Center, Prestige BioPharma Korea, Busan 46726, Korea

**Keywords:** PAUF, toll-like receptors, TLR4, pancreatic cancer, MyD88, NF-κB, PD-L1

## Abstract

PAUF, a tumor-promoting protein secreted by cancer cells, exerts paracrine effects on immune cells through TLR4 receptors expressed on immune cell surfaces. This study aimed to investigate if PAUF elicits autocrine effects on pancreatic cancer (PC) cells through TLR4, a receptor that is overexpressed on PC cells. In this study, TLR4 expression was detected in PC cells only, but not normal pancreatic cells. The migration of TLR4 high-expressing PC cells (i.e., BxPC-3) was reduced by a selective TLR4 inhibitor, in a dose-dependent manner. Using TLR4 overexpressed and knockout PC cell lines, we observed direct PAUF-TLR4 binding on the PC cell surfaces, and that PAUF-induced cancer migration may be mediated exclusively through the TLR4 receptor. Further experiments showed that PAUF signaling was passed down through the TLR4/MyD88 pathway without the involvement of the TLR4/TRIF pathway. TLR4 knockout also downregulated PC membrane PD-L1 expression, which was not influenced by PAUF. To the best of our knowledge, TLR4 is the first receptor identified on cancer cells that mediates PAUF’s migration-promoting effect. The results of this study enhanced our understanding of the mechanism of PAUF-induced tumor-promoting effects and suggests that TLR4 expression on cancer cells may be an important biomarker for anti-PAUF treatment.

## 1. Introduction

Pancreatic cancer (PC), which mainly consists of pancreatic ductal adenocarcinoma (PDAC), is a fatal disease. Despite continuing efforts to develop effective treatments, the 5-year survival rate of PC remains at only 10%. The high mortality is notably attributed to its high metastatic capacity [1].

We previously reported that pancreatic adenocarcinoma up-regulated factor (PAUF) is a novel secreted protein that is substantially expressed in PDAC cells [2]. PAUF can induce PC progression by acting as a tumor microenvironment (TME) modulator in a paracrine manner, e.g., it enhances the immunosuppressive function of immune cells via TLR-mediated signaling pathways (TLR2 and TLR4) [3]. PAUF also induces tumor-promoting effects in an autocrine manner [4], but the PAUF binding receptor on cancer cells has yet to be identified. In ovarian cancer patients, a statistically significant but weak correlation between the protein expression of PAUF and TLR4 was reported [5], how the two molecules are related and whether TLR4 acts as a PAUF receptor have not yet been studied.

Toll-like receptors (TLRs) are pattern-recognition receptors which are mainly expressed on immune cells and play critical roles in innate immunity. TLRs are also expressed on many types of cancer cells and the activation of TLR signaling pathways can induce cancer proliferation, invasion, survival, and metastasis [6]. Of the numerous TLR subtypes, TLR4 is overexpressed on PDAC and its expression correlates with cancer invasiveness [7,8].

This study aimed to investigate the relationship between PAUF and TLR4 expressed on PC cells and their effects on cancer mobility. We tested the effects of PAUF addition or neutralization on the migratory and invasive ability of PC cells with high- or low-level of TLR4 expression. PAUF addition and neutralization were achieved using recombinant PAUF (rPAUF) and anti-PAUF antibody (α-PAUF), respectively. PC cells with high- and low-level of TLR4 expression were accomplished by creating cell lines with TLR4 overexpression (TLR4^OE^) and knockout (TLR4^KO^), respectively. We also examined if PAUF can bind directly to TLR4 expressed on the surface of PC cells, and if it can activate TLR4 signaling pathways. We expect this study to improve our knowledge of how the PAUF-induced signal transduction is initiated and what factors in cancer cells determine PAUF’s tumor promoting effects.

## 2. Results

### 2.1. TLR4 Was Expressed in PC Cells but Not in Normal Pancreatic Cells

Using RT-qPCR and Western blot, we showed that TLR4 is expressed in all the PC cell lines, at various levels, but not in the normal pancreatic cell line, and the expression of other TLR subtypes also differed significantly between different PC cell lines.

The mRNA and protein expression profiles of TLR subtypes in six PC cell lines (MiaPaCa-2, Panc-1, CFPAC-1, Capan-1, AsPC-1, BxPC-3) and a normal pancreatic cell line (HPDE) are presented in Figure 1A and Figure 1B, respectively. Meanwhile, the rank of TLR4 mRNA expression levels in the six PC cell lines from the current study was compared to that of the Cancer Cell Line Encyclopedia (CCLE). Relative mRNA expression levels of all the major TLRs were compared between Panc-1 and BxPC-3 PC cells, two cells representing TLR4 low- and high-expressing cells, respectively (mRNA levels were adjusted based on the two cell lines’ TLR4 data presented in Figure 1A). The comparison showed that besides TLR4, three other TLRs (TLR2, TLR3, and TLR6) were also significantly different in mRNA expression between the two cell lines (Figure 1C). This is the reason we engineered the TLR4 overexpressed Panc-1 (Panc-1_TLR4^OE^) and BxPC-3 TLR4 knockout (BxPC-3_TLR4^KO^) PC cells to exclude the interference from other TLR receptors.

### 2.2. Endogenous Ligands Induced PC Cell Migration via TLR4

Using TAK-242, a chemical TLR4 inhibitor, we showed that PC cell-secreted ligands induced PC migration via TLR4 by transwell assay.

BxPC-3 and Panc-1 cells were seeded in the upper chambers containing TAK-242 (at 1, 2, and 4 µM), and incubated for 24 h. As shown in Figure 2A, TAK-242 reduced the migration of BxPC-3 cells in a dose-dependent manner (Jonckheere-Terpstra test, *p* < 0.001). By contrast, the migration of Panc-1 cells was not changed by TAK-242. To confirm whether the reduction of migratory ability was owed to the cytotoxicity of TAK-242, we performed WST-1 assays. Cell viability was not changed after BxPC-3 and Panc-1 cells were treated by TAK-242 (at 1, 2, or 4 µM) for 24 h (Figure 2B).

### 2.3. Successful Generation of TLR4 Overexpressed and Knockout PC Cells

At mRNA and protein expression levels, we confirmed TLR4 overexpression and knockout in four engineered stable PC cell lines. And we found that TLR4 expression does not impact PAUF concentration, but PAUF concentration tends to upregulate TLR4.

We confirmed TLR4 overexpression in the Panc-1_TLR4^OE^ cells, compared to the Panc-1_Mock cells, using immunofluorescence assay (Figure 3A), flow cytometry (Figure 3B) and Western blot (Figure 3C). We confirmed CRISPR/Cas9 knockout of TLR4 in BxPC-3 cells as follows: first, lentiviral transduction of scrambled sgRNA (NTC) and TLR4 sgRNAs (sgRNA1, sgRNA2, and sgRNA3) was evaluated by Cas9 mRNA expression (Figure 3D). Next, from a total of 32 single colonies isolated after sgRNA transduction, seven clones with loss-of-function (LoF) TLR4 were selected and pooled to generate the BxPC-3_TLR4^KO^ cells (Figure 3E), and the TLR4 expression in BxPC-3_NTC and BxPC-3_TLR4^KO^ pool was compared and shown in Figure 3C.

In the PC cell lines from the CCLE database (n = 52), there is a weak positive correlation (*r* = 0.363, *p* = 0.008) between TLR4 and PAUF mRNA expression (Figure 3F). To find out if TLR4 expression may impact PAUF expression, we measured PAUF concentration in these engineered cell lines. PAUF expression level was not changed by TLR4 overexpression or knockout (Figure 3G). Meanwhile, to examine if rPAUF may influence TLR4 expression, Panc-1 and BxPC-3 cells were treated by rPAUF (0, 0.1, 1, 3 μg/mL) for 6 h, and mRNA was extracted for RT-qPCR. As shown in Figure 3H, rPAUF (at 3 μg/mL) mildly increased the TLR4 mRNA expression in Panc-1 (*p* < 0.05) but not in BxPC-3, no dose-dependency was observed. As a positive control [7], shown in Figure 3I, lipopolysaccharide (LPS) notably increased TLR4 mRNA expression in the BxPC-3 cells, in a dose-dependent manner (*p* = 0.004).

### 2.4. PAUF Induced PC Cell Migration Was Dependent on TLR4 Expression

Using cell migration/invasion assays, we observed that PAUF was able to induce PC cell migration and invasion, but only in cells with high expression of normal function TLR4.

First, we investigated the impacts of low- and high-level of PAUF (using rPAUF and α-PAUF) on the migration of PC cells with TLR4 overexpression (Panc-1_TLR4^OE^) or knockout (BxPC-3_TLR4^KO^).

As shown in Figure 4A, a nearly 50% higher cell migration was observed in the Panc-1_TLR4^OE^ cells than in the Panc-1_Mock cells (*p* < 0.05), and rPAUF further increased the cell migration in the TLR4^OE^ cells by another 50% (*p* < 0.0001), but not in the Mock cells. By contrast, approximately 50% lower cell migration was observed in the BxPC-3_TLR4^KO^ cells than in the BxPC-3_NTC cells (*p* < 0.0001), and rPAUF significantly increased the cell migration in the NTC cells by approximately 30% (*p* < 0.001), but not in the TLR4^KO^ cells.

As shown in Figure 4B, when the PC cells were treated with α-PAUF, the migration of Panc-1_Mock cells was not changed, but the increased migration resulted from TLR4 overexpression was almost completely offset in the Panc-1_TLR4^OE^ cells (*p* < 0.05). In the BxPC-3_NTC cells, reduced migration achieved by α-PAUF was similar to that achieved through TLR4 knockout (*p* < 0.0001), and α-PAUF did not further reduce the migration of the TLR4^KO^ cells.

Similar results were observed in invasion assays. As shown in Figure 4C, rPAUF increased the invasive ability of Panc-1_TLR4^OE^ cells, but not in Panc-1_Mock cells. And α-PAUF antibody reduced the invasive ability of Panc-1_TLR4^OE^ cells, but not in Panc-1_Mock cells.

To confirm the involvement of TLR4 in the migration of PC cells, we used a selective TLR4 inhibitor, TAK-242 (Figure 4D) to eliminate the function of TLR4. In the cells of low-level TLR4 expression (Panc-1_Mock and BxPC-3_TLR4^KO^), TAK-242 did not impact the cell migration, regardless of rPAUF treatment. In the cells of high-level TLR4 expression (Panc-1_TLR4^OE^ and BxPC-3_NTC), TAK-242 reduced basal migration (by about 20% and 40%), and the increased migration due to rPAUF exposure (by about 25% and 50%) was completely offset by TAK-242.

### 2.5. PAUF Bound to TLR4 on the Surface of Pancreatic Cancer Cells

Using Western blot and immunofluorescence assay, we observed direct binding of PAUF and TLR4 on the PC cell surface.

Our previous research revealed that PAUF binds to TLR4 proteins on myeloid-derived suppressor cells (MDSCs) and induces immune suppression. To investigate the potential binding of PAUF with TLR4 on the plasma membrane of the cancer cells, we used a chemical crosslinker BS^3^, which is a membrane-impermeable agent that possesses a spacer arm of 1.14 nm. Panc-1_Mock and Panc-1_TLR4^OE^ cells were treated by 2 μg/mL rPAUF or 2.5 mM BS^3^ alone, or in combination. A band of approximately 100 kDa was present by Western blot in all Panc-1_TLR4^OE^ cells but not in Panc-1_Mock cells, regardless of rPAUF and BS^3^ treatment, which was estimated to be a TLR4 monomer. Furthermore, only when the Panc-1_TLR4^OE^ cells were treated by both rPAUF and BS^3^, an additional band of high molecular mass (≥300 kDa) was also detected, which is estimated to be a TLR4 complex consisting of TLR4 and PAUF (Figure 5A).

To confirm whether PAUF can bind directly to TLR4 on the surface of cancer cells, we performed proximity ligation assay (PLA). A schematic illustration of PLA is shown in Figure 5B. Panc-1_Mock and Panc-1_TLR4^OE^ cells were treated with crosslinker BS^3^, with or without rPAUF. As expected, BS^3^-treated Panc-1_Mock cells did not show any signal, regardless of rPAUF treatment. In comparison, BS^3^-treated Panc-1_TLR4^OE^ cells showed a strong signal on the plasma membranes when treated with rPAUF (Figure 5C).

### 2.6. PAUF Activated TLR4 through MyD88-Dependent Signaling Pathway

Using immunoprecipitation, Western blot, and reporter gene assays, we discovered that PAUF activated MyD88 but not TRIF of TLR4 downstream signaling pathway.

Immunoprecipitation (IP) assays were conducted to detect the binding of adaptor proteins MyD88 and TRIF to TLR4, respectively. Panc-1_Mock and Panc-1_TLR4^OE^ cells were treated with rPAUF for different time periods and then IP was performed to the cell lysates using α-MyD88 or α-TRIF antibody. Immunoprecipitated proteins were then subject to Western blot to detect TLR4 and MyD88/TRIF. We observed binding of MyD88 to TLR4 only 10 min after rPAUF (0.1 μg/mL) addition (Figure 6A). However, rPAUF did not affect the binding of TRIF to TLR4 (Figure 6B). Western blot against TLR4 and MyD88/TRIF in the whole cell lysates was used as references for TLR4 binding in the IP assays.

To confirm that PAUF signals were mediated via the TLR4/MyD88 pathway, not the TLR4/TRIF pathway, we performed dual-luciferase and Western blot analyses. For the dual-luciferase assay, Panc-1 cells were transiently transfected with NF-κB-luc, pRL-TK, and TLR4 expression vectors. TLR4 protein expression was confirmed in the Panc-1_trans TLR4^OE^ cells by a Western blot analysis (Figure 6C). As shown in Figure 6D, in the Panc-1_trans Ctrl cells, rPAUF did not have any impact on the luciferase activity. However, the NF-κB luciferase activity was reduced by 40% (*p* < 0.0001) by an inhibitor of IKK XII (IKK is a common upstream signaling factor of NF-κB/P65 in the TLR4 pathway). On the other hand, after Panc-1 cells were transiently overexpressed with TLR4, the level of luciferase activity was increased by 3.8 folds (*p* < 0.0001), and rPAUF treatment further increased the luciferase activity by 4.6 folds (*p* < 0.0001) in the TLR4 transiently overexpressed cells (trans TLR4^OE^), compared to that of the Panc-1_trans Ctrl cells. However, the NF-κB luciferase activity was reduced by 30% (*p* < 0.0001) by the IKK inhibitor, and the additional increase in luciferase activity induced by rPAUF in the trans TLR4^OE^ cell was completely offset. To investigate whether downstream components of TRIF pathway are activated by PAUF, Western blot analyses were conducted. As shown in Figure 6E, after the Panc-1_Mock and Panc-1_TLR4^OE^ cells were incubated with rPAUF from 10 min to 4 h, rPAUF did not significantly impact phosphorylation of TBK1 and IRF3 in either cell line.

### 2.7. PAUF Up-Regulated Programmed Death-Ligand 1 (PD-L1) Expression in Cancer Cell Cytoplasm

After we confirmed that PAUF selectively simulated TLR4/MyD88 signaling pathway, we further investigated if PAUF can up-regulate PD-L1 expression, because a recent study showed that LPS can mediate PD-L1 upregulation in PDAC via the same pathway [11].

Using flow cytometry analyses and RT-qPCR, we discovered that similar to LPS, PAUF caused an increase in overall PD-L1 expression in PC cells, but neither LPS nor PAUF increased PD-L1 expression on PC cell surface.

PD-L1 expression levels in the PC cells with low- or high-level of TLR4 expression were evaluated after the cells were treated by rPAUF (0, 0.1, 3 μg/mL). In this study, LPS (0, 1, 5 μg/mL) was used as a positive control for rPAUF [11]. As shown in Appendix A, PD-L1 was expressed on the surface of Panc-1 cells at a very low level (1.05), in comparison to BxPC-3 cells (2.10), and rPAUF did not show any significant effect on the PD-L1 expressed on cell surface, but the knockout of TLR4 in BxPC-3 cells led to a marked decrease in PD-L1 expression (from 2.10 to 1.67). LPS did not show any effect on the PD-L1 expression on Panc-1_Mock or Panc-1_TLR4^OE^ cell surface (Appendix A). Contrasting the results of PD-L1 expressed on cell surface, PD-L1 expressed by the whole cell was significantly up-regulated by both rPAUF and LPS treatments, while at the lower concentration of both treatments, the responses were greater in BxPC-3 cells than in the Panc-1 cells (Appendix A).

## 3. Discussion

PAUF has been extensively studied for the past decade owing to its versatile tumor-promoting effects. However, not until this study has PAUF’s receptor on cancer cell surfaces been clearly identified. This is partly because cancer cell surfaces are known to express a staggering amount of antigen receptors [12], many of which share common intracellular signaling pathways with PAUF such as JNK, ERK, and Wnt/β-catenin [4,13]. The current study not only shows that PAUF can bind directly to TLR4 on pancreatic cancer (PC) cell surfaces and promote cell migration exclusively through the MyD88-dependent pathway. It also suggests that cancer TLR4 expression may be a critical biomarker in anti-PAUF treatment.

In this study, we did not detect TLR4 expression in the HPDE normal pancreatic cells (either mRNA or protein), contrasting the positive findings in all six PC cell lines. However, TLR4 expression levels varied substantially among these cell lines. To assess the reliability of the mRNA expression data in the current study, we compared our data to those retrieved from the Cancer Cell Line Encyclopedia (CCLE) and found agreement between them. Except that the rank of TLR4 mRNA expression of CFPAC-1 of the six PC cell lines was 4th in our study but 3rd in CCLE, the rank of TLR4 protein expression is however, consistent with the CCLE finding. The discordance of mRNA and protein expression was frequently observed in cancer cells [14,15]. The vastly varied TLR4 expression was expected as TLR4 plays multiple roles in shaping the tumor microenvironment [16], and the latter is famously known of its complexity.

In addition, we observed significant differences between different PC cell lines not only in TLR4, but also in other TLRs (e.g., TLR2, TLR3, and TLR6). This is the reason we engineered the TLR4 overexpressed Panc-1 (Panc-1_TLR4^OE^) and BxPC-3 TLR4 knockout (BxPC-3_TLR4^KO^) PC cells, to exclude the interference from other TLR receptors.

There is increasing evidence supporting the association of exogenous and endogenous substances with TLR mediated cancer migration and invasiveness [7,8,17]. To find out if endogenous substances including PAUF [2] can exert a cancer migration-promoting effect via TLR4, we conducted a migration study in PC cell lines with low to high level of TLR4 activity, without adding any migration-promoting treatment. The hierarchy of TLR4 activity was achieved using 0 to 4 μM of TAK-242, a small molecule that selectively inhibits TLR4 expressed on cell surfaces [18]. This experiment showed a dose-dependent suppressive effect of TAK-242 on the migration of TLR4 high-expressing BxPC-3 cells, but not in the TLR4 low-expressing Panc-1 cells. Notably, the TAK-242 concentrations we used were not cytotoxic to the PC cells, therefore the suppression of BxPC-3 migration is likely to be mainly, if not solely, owing to the block of TLR4 signaling pathway. This study confirms that TLR4 signal transduction can be induced by PC cell originated endogenous ligand(s). Meanwhile, the lack of impact of TAK-242 on the migration of TLR4 low-expressing Panc-1 cells is in agreement with a previous study. In that study, TAK-242-attenuated prostate cancer cell migration was shown to be dependent on ERG (transcription factor) activated TLR4 gene expression [19].

Next, we carried out different experiments to discern if PAUF is among the endogenous substances that promote PC cell migration via the TLR4 pathway.

The first experiment observed significantly altered PC cell migration/invasion followed by artificial manipulation of PAUF using rPAUF or α-PAUF, and these changes were seen only in those TLR4 high-expressing PC cells but not in those TLR4 low-expressing PC cells. Consistent findings were seen in a study using the same PC cells with TAK-242 deprived TLR4 activity. These results unambiguously suggests that PAUF’s impact on PC cell migration are dependent on TLR4 expression. The monoclonal anti-PAUF antibody used in this study is currently at phase I clinical trial for safety evaluation (clinicaltrials.gov identifier: NCT05141149). However, our findings imply a potential clinical application of using TLR4 expression level as a biomarker to identify anti-PAUF treatment responders in later phase clinical trials.

Cancer migration/invasion-promoting effects mediated by TLR4 pathways were observed on molecular levels (such as E-cadherin and MMP9) in multiple studies [8,17], for study simplicity, we did not evaluate migration-related molecular changes in this study.

Evidence supporting the finding that PAUF’s effects are mediated exclusively through TLR4 signaling pathway includes: (1) In PC cells lacking TLR4 expression or activity, rPAUF shows no migration-promoting effect at all, and α-PAUF also lost its migration-blocking effect completely (Figure 4). (2) The 50% increase and 50% decrease in migratory ability of the Panc-1_TLR4^OE^ and BxPC-3_TLR4^KO^ cell lines respectively (compared to the control cells) were seemingly caused by other TLR4 ligands rather than PAUF, because the PAUF concentrations were not influenced by TLR4 overexpression or knockout (Figure 3F). However, neutralizing PAUF in the Panc-1_TLR4^OE^ cell line completely offset the increased migration. Likewise, neutralizing PAUF in the BxPC-3_NTC cell line led to a 50% migration reduction, similar to that caused by TLR4 knockout. And neutralizing PAUF together with TLR4 knockout did not produce any greater migration reduction, ruling out any significant effects caused by other ligands (Figure 4B).

In the second experiment, we observed a weak positive correlation between TLR4 and PAUF mRNA expression using the PC cell data from the CCLE database (n = 52, *r* = 0.363, *p* = 0.008), a similar correlation was reported in ovarian cancer patients at the protein level (n = 182, *r* = 0.256, *p* = 0.001) [5]. In order to find out if this correlation was caused by PAUF exposure, we conducted an experiment in two PC cell lines and found out that rPAUF (at 3 μg/mL) mildly up-regulated TLR4 expression in Panc-1 cells. A previous study showed that the well-known TLR4 ligand LPS induces TLR4 protein expression [7], in the current study, LPS also induced TLR4 mRNA expression, but the trend is only significant in BxPC-3 cells not in Panc-1 cells (Figure 3I). We did not observe upregulation of TLR4 by rPAUF in BxPC-3 cells though, probably because mRNA expression results do not always correlate well with protein expression [14,15].

In the next experiment, we obtained evidence for the direct binding of PAUF to TLR4, using a proximity ligation assay (PLA) in rPAUF treated Panc-1_TLR4^OE^ cells. Notably, a very small amount of PAUF-TLR4 binding was observed in these cells even without addition of rPAUF. We estimated that it could be from the non-specific binding of primary antibodies (polyclonal antibodies) to other proteins on the PC cell surfaces. And the signals are unlikely caused by endogenous PAUF because we already show that Panc-1_Mock and Panc-1_TLR4^OE^ cells have similar PAUF secretion (Figure 3G), nor were the signals possibly detected in cytoplasm even though both PAUF and TLR4 also expressed there [20], because the crosslinker BS^3^, the primary and secondary antibodies used for PLA were unable to penetrate cell membrane.

After the PAUF-TLR4 relationship was confirmed, we sought to find out which TLR4 downstream adaptors were specifically reacting to the PAUF signal. In the Panc-1_TLR4^OE^ cells, a rapid binding of adaptor MyD88 to TLR4 was observed as early as 10 min after the rPAUF treatment, while the binding of another adaptor TRIF to TLR4 appeared to be independent from rPAUF treatment. A Western blot analysis showed no phosphorylation change in TRIF downstream components (TBK1 and IRF3) after the rPAUF stimulation. In addition, a reporter gene assay showed significantly up-regulated NF-κB activity after the rPAUF exposure in transient TLR4 overexpressed Panc-1 cells, and the activation was drastically reduced by IKK selective inhibitor XII. These findings reinforced that PAUF-induced TLR4 signaling is likely passed through the MyD88/NF-κB pathway, but unlikely through the TRIF/IRF3 pathway. While our study is consistent with previous studies that investigated TLR4 expressed on cancer cells which report the involvement of TLR4/MyD88 in tumor progression [7,11,17], the involvement of the TLR4/TRIF pathway in tumor progression is not well-supported in the literature. Clarification of the specific pathway is important because it gives us better mechanistic understanding of PAUF’s effects and enables us to conceive new treatment strategies. For example, a bispecific antibody that targets both PAUF and B7-H3 (also known as CD276) may generate a tumor specific but more potent anti-tumor effect. A recent study showed that B7-H3 is a checkpoint molecule that is highly-expressed on PC cells and known to increase the invasiveness of PC cells through the TLR4/NF-κB pathway [21]

Continually, there has been a lot of attention on another immune checkpoint, PD-L1. A previous study reported 10 μg/mL of LPS upregulated PD-L1 expression in Panc-1 cells via the TLR4/MyD88/NF-κB signaling pathway [11]. However, in our study (Appendix A), the same concentration of LPS did not cause any change in PD-L1 expression on Panc-1 cell surfaces, neither did rPAUF show any such effect. Meanwhile, the TLR4 knockout showed a large impact on the PD-L1 expression on BxPC-3 cells, but TLR4 overexpression did not impact PD-L1 expression in Panc-1 cells at all. The discrepancy between our and previous studies and the different results between BxPC-3 and Panc-1 cells may be because we measured PD-L1 expression on PC cell surfaces but not in the cell lysates. It is known that PD-L1 is sparse on the surface of Panc-1 but expressed at relatively high levels on BxPC-3 cells [22,23], while in the cytoplasm, both cell lines show high levels of PD-L1 expression [23]. Since cancer cell membrane-expressed PD-L1, not PD-L1 in cytoplasm, has biological significance [24], and is related to PD-L1 inhibitor treatment outcomes [25], our study shows that PAUF is unlikely to impact the PD-1 and PD-L1 treatment outcomes, even if it can change PD-L1 level in cytoplasm via TLR4.

Despite the findings mentioned above, more questions emerged and await answers: (1) TLR4 was identified as the receptor that exclusively mediates PAUF’s migration-promoting effects, but the receptors that mediate PAUF’s other tumor-promoting effects still remain unclear (e.g., PAUF’s proliferation-promoting effects via the Wnt/β-catenin pathway [13]). (2) Even in those TLR4 high-expressing PC cells (e.g., BxPC-3), up to 50% of the endogenous ligands-induced cell migration appeared to be uninfluenced by TLR4 knockout (Figure 4B), suggesting involvements of other pathways, possibly HGF/c-MET [26] and CD44 [27], which are well known to play important roles in cancer metastasis.

Animal studies are now underway to find out how tumor TLR4 expression levels may impact the anti-tumor efficacy of the anti-PAUF treatment, and to explore the synergistic anti-tumor efficacy of inhibition on both TLR4/MyD88 dependent PAUF and B7-H3.

## 4. Materials and Methods

### 4.1. Cells

The normal human pancreatic cell line HPDE and human pancreatic cancer cell line Capan-1 were obtained from Korea Research Institute of Bioscience and Biotechnology (KRIBB). The MiaPaCa-2, Panc-1, BxPC-3, AsPC-1, and CFPAC-1 cell lines were purchased from American Type Culture Collection (ATCC, Manassas, VA, USA). The HEK293T cell line was a gift from Professor Dae-Sik Lim of the Korea Advanced Institute of Science and Technology.

The HPDE cell line was cultured in the defined Keratinocyte SFM (K-SFM) medium (Gibco, CA, USA). Capan-1 and CFPAC-1 cell lines were maintained in Iscove’s Modified Dulbecco’s Medium (IMDM, WELGENE, Daegu, Gyeongsangbuk-do, Korea). MiaPaCa-2 and Panc-1 cell lines were cultured in Dulbecco’s Minimal Essential Medium (DMEM, Cytiva, Marlborough, MA, USA) and BxPC-3 and AsPC-1 cell lines were maintained in RPMI-1640 (WELGENE). All mediums were supplemented with 10% fetal bovine serum (FBS, Cytiva) and 1% penicillin/streptomycin (Gibco) at 37 °C with 5% CO_2_.

### 4.2. Reagents

Recombinant PAUF (rPAUF) protein was prepared as previously described [4]. Humanized anti-PAUF neutralizing monoclonal antibody was prepared as described in the Patent Cooperation Treaty (PCT) WO2019022281A1. Human IgG isotype control (cat# 31154, Thermo Fisher Scientific, Waltham, MA, USA) was used as a control antibody for the anti-PAUF antibody. Antibodies against TLR2 (cat# AF2616), TLR4 (cat# AF1478), and Goat IgG-phycoerythrin (IgG-PE, cat# F0107) were purchased from R&D Systems, Inc (Minneapolis, MN, USA). eFluor 450 conjugated mouse anti-PDL1 antibody (cat# 48-5983-42) and eFluor 450 conjugated mouse IgG1, kappa isotype control (cat# 48-4714-82) from Invitrogen (Waltham, MA, USA) was used for detection of surface PD-L1 expression in PC cells. Goat IgG isotype control (cat# AB-108-C) and antibodies against MyD88 (cat# 4283S) were obtained from Cell Signaling Technology (Danvers, MA, USA). Antibodies against β-actin (cat# sc-47778) and GAPDH (cat# sc-47724) were purchased from Santa Cruz Biotechnology (Dallas, TX, USA). Anti-TRIF antibody (cat# NB120-13810) was obtained from Novus Biologicals (Centennial, CO, USA). Streptavidin-HRP (cat# 21130) was purchased from Thermo Fisher Scientific. TLR4 inhibitor TAK-242 (cat# 13871) was purchased from Cayman Chemical (Ann Arbor, MI, USA). For migration and invasion assays, Giemsa solution (cat# 32884, Sigma-Aldrich, St. Louis, MO, USA) and Matrigel (cat#354234, Corning Inc., Corning, NY, USA) were used. For determination of protein concentration, the Pierce^TM^ BCA Protein Assay Kit (cat# 23225, Thermo Fisher Scientific) was used. For protein detection in Western blot analyses, the ECL^TM^ reagents (cat# K-12045-D50, Advansta, San Jose, CA, USA) was used. Human TLR4 open reading frame (cat# HG10146-M) was obtained from Sino Biological, Inc., (Beijing, China). For luciferase reporter assays, the pGL4.32[luc2P/NF-κB-RE/Hygro] and pRL-TK vectors, Lipofectamine 2000 transfection reagent (cat# 11668019) was purchased from Invitrogen, and Dual-Luciferase Reporter Assay System (cat# E1960) were purchased from Promega (Madison, WI, USA), the pDUO-mcs vector (cat# pduo-mcs) was purchased from InvivoGen (San Diego, CA, USA), and the IKK inhibitor XII (cat# 401491) were purchased from Sigma-Aldrich. For proximity ligation assays (PLA), the BS^3^ crosslinker (cat# A39266) was purchased from Thermo Fisher Scientific, Duolink In Situ PLA probe anti-rabbit MINUS (cat# DUO92005-30RXN), Duolink In Situ PLA Probe anti-Goat PLUS (cat# DUO92003), Duolink in situ detection reagents green (cat# DUO92014), Duolink in situ wash buffers (cat# DUO82049-4L), and Duolink in situ mounting medium with DAPI (cat# DUO82040) were purchased from Sigma-Aldrich. Vectashield antifade mounding medium (cat# H1900) was purchased from Vector Laboratories Inc., (Newark, CA, USA).

### 4.3. RNA Isolation and RT-qPCR

Total RNA was extracted using Trizol (FAVORGEN, Vienna, Austria) according to the manufacturer’s instructions. Five hundred nanograms of cellular RNA was converted to cDNA using PrimeScript^TM^ RT Master Mix (Takara Bio, Shiga, Japan) in a total volume of 20 µL. RT-qPCR analysis was performed using an AriaMx Real-Time PCR System (Agilent Technologies, Santa Clara, CA, USA). The primer sequences of the genes for analysis are listed in Table 1. Gene expression levels were normalized to those of the GAPDH.

### 4.4. Cell Migration and Invasion Assays

Cell migration and invasion were determined using a 24-well Transwell^®^ system with a pore size of 8 µm and a non-coated membrane (Corning). For invasion assay, upper chamber of transwell was coated with 300 μg/mL Matrigel.

The lower chamber of the transwell was filled with 700 µL culture medium (supplemented with 10% FBS). A total of 5 × 10^4^ BxPC-3 cells or 8 × 10^4^ Panc-1 cells were seeded into the upper well in 200 µL serum-free medium containing 1, 2, or 4 µM TAK-242 and incubated for 24 h at 37 °C. Migrated cells were fixed to the underside with absolute methanol and then stained with Giemsa solution. The non-migrated cells on the upper side of the membrane were removed by cotton swabs. Images of the migrated cells were captured using an inverted microscope and the number of cells were counted using the ImageJ software (Version 1.53, Wayne Rasband et al., USA) from six randomly selected fields for each condition. The number of migrated cells was presented as the percentage of migration relative to that of the cells incubated without TAK-242, which was considered 100%.

To determine the effect of PAUF and TLR4 interaction on PC cell migration, PC cells were seeded in the upper well with 0.5 µg/mL rPAUF or 20 µg/mL α-PAUF (a concentration estimated to be sufficient to maximally neutralize PAUF protein) and incubated for 24 h at 37 °C. After fixation and staining, the non-migrated cells on the upper side of the membrane were removed by cotton swabs. The numbers of cells were visualized and counted from six randomly selected fields using the ImageJ software. The migratory and invasive ability were represented as the percentage of migrated cells relative to that of PC cells (Panc-1_Mock or BxPC-3_NTC) incubated without rPAUF or α-PAUF, which was used as controls with the migratory or invasive ability of 100%.

### 4.5. Cell Viability Assay

Cell viability was determined using WST-1 reagent (Sigma-Aldrich) according to the manufacturer’s instructions. Briefly, BxPC-3 and Panc-1 cells were seeded in 96-well plates at a density of 5 × 10^4^ and 8 × 10^4^ cells/well, respectively, and cultured with TAK-242 (1, 2, or 4 μM) for 24 h. After that, 20 µL WST-1 was added to each well and incubated for 1 h at 37 °C. The absorbance at 450 nm was detected using an ultraviolet spectrophotometer (Versamax, Molecular Devices, San Jose, CA, USA).

### 4.6. Lentiviral-Mediated TLR4 Overexpression in Panc-1 Cells

To construct the TLR4 overexpression (TLR4^OE^) lentiviral vector, human TLR4 open reading frame was subcloned into the pLVX-EF1α-IRES-Puro vector (Addgene, Watertown, MA, USA) between the SpeI and BamHI sites. For the generation of lentiviral particles, HEK293T cells were transfected with 12 µg TLR4^OE^ lentiviral vector, 9 µg psPAX2 packaging vector (Addgene), and 3 µg pMD2.G envelope vector (Addgene) using 36 µg polyethylenimine (PEI, Polysciences, Warrington, PA, USA). After 48-h incubation, the culture supernatants were harvested and concentrated using the Lenti-X concentrator (Takara Bio). Concentrated viral particles were stored at −80 °C and thawed immediately before titration and transduction.

For lentivirus titration, Panc-1 cells were infected with 2, 20, or 200 µL lentiviral particles in the presence of 8 µg/mL polybrene. After 24-h incubation, the genomic DNA of the cells was isolated using PureLink^TM^ Genomic DNA Mini Kit (Thermo Fisher Scientific) and subjected to qPCR for detection of both WPRE in the lentiviral vector, and human albumin as reference gene. The following primer sequences were used to detect WPRE: 5′-GGCTGTTGGGCACTGACA-3′ and 5′-CCGAAGGGACGTAGCAGAA-3′. The following primer sequences were used to detect human albumin: 5′-GTCATCTCTTGTGGGCTGTAATC-3′ and 5′-CTATCCAAACTCATGGGAGCTG-3′. The number of lentiviral vector copies per cell was calculated by normalizing the number of WPRE copies to the number of albumin copies.

For lentiviral transduction, Panc-1 cells were infected with control or TLR4^OE^ lentivirus with multiplicity of infection (MOI, pfu/cell) of 10 in the presence of 8 µg/mL polybrene for 24 h. Infected cells were selected by incubation with 4 µg/mL puromycin (Sigma-Aldrich) for 2 weeks, and then maintained in DMEM supplemented with 10% FBS. The addition of puromycin allowed for selection of cells that were stably overexpressing TLR4. Short tandem repeats (STR) fingerprinting was performed to confirm the identity of Panc-1_Mock and Panc-1_TLR4^OE^ cells.

### 4.7. TLR4 Knockout via CIRSPR-Cas9 in BxPC-3 Cells

Guide RNA sequences that target human TLR4 were designed at CRISPR design website (http://crispr.mit.edu/, accessed on 18 July 2022). The oligonucleotide sequences of the TLR4 sgRNAs and scrambled sgRNA area shown in Table 2. U6 promoter-TLR4 sgRNA-tracrRNA scaffold-EF-1α promoter sequences were synthesized by Integrated DNA Technologies (Coralville, IA, USA) and then subcloned into the lentiCRISPRv2-puro (Addgene) between the Acc65I and XbaI sites. For the generation of lentiviral particles, HEK293T cells were transfected with 12 µg TLR4 sgRNA lentiviral vector, 9 µg psPAX2 packaging vector, and 3 µg pMD2.G envelope vector using 36 µg PEI. The culture mediums were collected after 48-hr incubation and concentrated using the Lenti-X concentrator. Concentrated viral particles were stored at −80 °C and thawed immediately before titration and transduction. For lentivirus titration, the same procedures were conducted as described in the previous section.

Next, BxPC-3 cells were infected with scrambled sgRNA, TLR4 sgRNA1, sgRNA2, or sgRNA3 lentivirus at MOI of 7 in the presence of 8 µg/mL polybrene for 24 h. Infected cells were selected by incubation with 3 µg/mL puromycin for 2 weeks, and then maintained in RPMI-1640 supplemented with 10% FBS. Transduction of the virus was confirmed by RT-qPCR to detect the expression of Cas9. The primer sequences for Cas9 are shown in Table 1. Thirty-two colonies of BxPC-3 cells transfected with TLR4 sgRNA1, sgRNA2, or sgRNA3 were isolated after limiting dilution followed by colony expansion. Sanger sequencing was performed to detect mutations at the sgRNA target sites in the TLR4 gene and biallelic mutations were discovered in eight clones (Appendix A). TLR4 gene sequences of these eight clones were then translated into amino acid sequences using Vector NTI software (Thermo Fisher Scientific) to predict the functionality of TLR4. And seven clones showed loss-of-function (LoF) and one clone had unknown function of TLR4 (Appendix A). Western blot analyses confirmed the TLR4 knockout (TLR4^KO^) in these seven clones, thus they were mixed at equal amounts to generate BxPC-3 TLR4 knockout (TLR4^KO^) pool. BxPC-3 no template control (NTC) cells are cells infected with the control virus containing scrambled sgRNA. Short tandem repeats (STR) fingerprinting was performed to confirm the identity of BxPC-3_NTC and BxPC-3_TLR4^KO^ cells.

### 4.8. Immunofluorescence (IF) Assay

Panc-1_Mock and Panc-1_TLR4^OE^ cells were grown on glass coverslips, fixed in 4% paraformaldehyde, and incubated with anti-TLR4 antibody at 4 °C overnight. After washing with PBST (0.05% Tween 20 in PBS), cells were stained with PE-conjugated anti-goat secondary antibody. The nuclei were counterstained with DAPI (0.1 µg/mL) at 37°C for 10 min. Slides were mounted using Vectashield antifade mounting medium. Images were acquired with a NIKON Eclipse Ni microscope (Nikon, Tokyo, Japan).

### 4.9. Flow Cytometry

Panc-1_Mock and Panc-1_TLR4^OE^ cells were harvested and incubated with goat anti-TLR4 or goat IgG isotype on ice for 30 min. After washing, cells were stained with PE-conjugated anti-goat secondary antibody on ice for 30 min. Fluorescence-activated cell sorting (FACS) buffer (5% FBS in PBS) was used to wash cells and dilute all antibodies. Samples were measured and analyzed using a Novocyte flow cytometry system and NovoExpress software (ACEA Bioscience Inc., San Diego, CA, USA).

To determine the changes of the surface PD-L1 expression induced by rPAUF or LPS in TLR4 high- and low- expressing PC cells (BxPC-3_NTC, BxPC-3_TLR4^KO^, Panc-1_Mock, or Panc-1_TLR4^OE^), cells were incubated with serum-free media containing rPAUF (0, 0.1, 3 μg/mL) or LPS (0, 1, 5 μg/mL) for 24 h. Harvested cells were incubated with eFluor 450 conjugated anti-PD-L1 antibody or isotype control on ice for 30 min. After washing, cells were counted, and mean fluorescence intensity (MFI) were estimated.

### 4.10. Western Blot Analysis

Cells were rinsed in PBS and homogenized with protein extraction RIPA solution (50 mM Tris-Cl [pH 7.4], 150 mM NaCl, 1 mM PMSF, 0.1% SDS, 1% NP-40, 2 mM EDTA, 50 mM NaF, 0.5% sodium deoxycholate, and 1 mM Na_3_VO_4_). To determine the activation of the TLR4/MyD88 dependent signaling factors by PAUF, cells were incubated in serum-free media with or without 0.1 μg/mL rPAUF for different time periods. After preparation of cell lysate, the protein concentration was determined using the Pierce^TM^ BCA Protein Assay Kit. Protein samples at equal amounts were separated by SDS-PAGE and transferred to a nitrocellulose membrane (GE Healthcare, Chicago, IL, USA). The membranes were blocked with 5% non-fat dry milk and probed with the appropriate primary antibodies diluted in Tris-buffered saline with Tween-20 (TBST) containing 3% bovine serum albumin (BSA). The membranes were consequently incubated with secondary horseradish peroxidase-conjugated antibodies. The signals were detected by the Azure C300 gel imaging system (Azure Biosystems, Dublin, CA, USA).

To determine whether PAUF induces the formation of TLR4 complexes, Panc-1_Mock and Panc-1_TLR4^OE^ cells were harvested and incubated with or without 1 μg/mL rPAUF for 15 min at 37 °C. A chemical crosslinker BS_3_ (2.5 mM) was consequently treated for 30 min at room temperature. After neutralization by 20 mM Tris, cells were lysed using IP lysis buffer (50 mM Tris-Cl [pH 7.4], 150 mM NaCl, 1% NP-40, 2 mM EDTA) for 30 min at 4 °C. Individual proteins were separated by SDS-PAGE and transferred to a nitrocellulose membrane. The membranes were blocked and probed with an anti-human TLR4 antibody or anti-β-actin antibody. Membranes were incubated with peroxidase-conjugated secondary antibodies and then bands were detected by the Azure C300 gel imaging system.

### 4.11. PAUF Detection by ELISA

Panc-1_Mock, Panc-1_TLR4^OE^, BxPC-3_NTC, or BxPC-3_TLR4^KO^ cells were cultured in DMEM or RPMI-1640 medium for 48 h. The supernatants were concentrated by Amicon Ultra-15 Centrifugal Filter Units (Millipore, Burlington, MA, USA) and collected for PAUF detection by ELISA. Plates were coated with α-PAUF (5 μg/mL) overnight at room temperature and then incubated with the concentrated supernatants at 37 °C for 90 min. R4P-biotin detection antibodies (250 ng/mL, R4P is a rabbit anti-PAUF polyclonal antibody) were added and incubated at 37 °C for 90 min, and then streptavidin-HRP (1:5000) was added and incubated at 37 °C for 30 min. The PAUF expression level was detected at 450 nm using the Versamax ultraviolet spectrophotometer.

### 4.12. Proximity Ligation Assay (PLA)

Panc-1_Mock and Panc-1_TLR4^OE^ cells were grown on glass coverslips and treated with PBS or rPAUF (2 µg/mL) at 4 °C for 30 min. Cross-linker BS^3^ (2.5 mM) was added to the cells and incubated at 4 °C for 45 min. After neutralization, cells were fixed with 4% paraformaldehyde and blocked with Duolink Blocking Solution. The cells were incubated with primary antibodies that recognize PAUF and human TLR4 (α-PAUF and α-TLR4) at a concentration of 10 µg/mL overnight at 4 °C. After washing with Buffer A, cells were incubated with PLUS or MINUS PLA probe conjugated secondary antibodies. Cells were washed in Buffer A at room temperature for 10 min and then incubated with the ligase and ligase buffer at 37 °C for 30 min. Next, cells were washed again in Buffer A at room temperature for 10 min and incubated with the polymerase and amplification buffer at 37 °C for 100 min. Lastly, cells were washed in Buffer B for 20 min and then in 0.01X Buffer B at room temperature for 1 min. Duolink In Situ Mounting Medium with DAPI was added and samples images were acquired with a NIKON Eclipse Ni microscope.

### 4.13. Immunoprecipitation (IP) Assay

Panc-1_Mock and Panc-1_TLR4^OE^ cells were treated with rPAUF (0.1 µg/mL) for different time periods (0–120 min) and then protein samples were collected from the treated cells. The protein concentrations were determined using the Pierce^TM^ BCA Protein Assay Kit. 200 µg protein samples in IP lysis buffer were incubated with anti-MyD88 or anti-TRIF antibodies (α-MyD88 and α-TRIF) and then immunoprecipitated with Protein A agarose beads. Immunoprecipitated samples were subjected to SDS-PAGE gel electrophoresis and then transferred on a nitrocellulose membrane. After blocking using 5% non-fat dry milk, the membranes were probed with primary antibodies (1:500) overnight at 4 °C and the peroxidase-conjugated secondary antibodies (1:5000) at room temperature for 1 h. The blotting signal was then detected by ECL^TM^ reagents.

### 4.14. Luciferase Reporter Assays

Panc-1 cells were seeded in 60-mm dishes at a density of 1.5 × 10^5^ cells and cultured for 2 days. pGL4.32[luc2P/NF-κB-RE/Hygro] and pRL-TK vectors, together with TLR4 expression plasmid (vTLR4) or control plasmid (vCtrl) were transiently transfected into the Panc-1 cells to generate transient TLR4 overexpressed cells (Panc-1_trans TLR4^OE^) and the control cells (Panc-1_trans Ctrl), using the Lipofectamine 2000 transfection reagent. The vTLR4 was synthesized through inserting the TLR4 gene fragment into the pDUO-mcs vector, and the pDUO-mcs was used as the vCtrl. TLR4 gene fragment was obtained from AgeI and AvrII enzyme digestion of the PCR product of the TLR4 open reading frame (ORF) region (forward primer 5′-AAAAACCGGT ATGATGTCTGCCTCGCGCCT-3′, reverse primer 5′-ACCCCTAGGTCAGATAGATGTTGCTTCCTGCC-3′). After transfection for 3 h and further stabilization, the cells were treated with IKK inhibitor XII for 4 h and then stimulated by rPAUF (0.1 µg/mL) for 14 h. Luciferase activities were examined using a Dual-Luciferase Reporter Assay System according to the manufacturer’s instructions.

### 4.15. Statistical Analysis

All quantifiable experiment results were presented as mean ± standard deviation (SD) from 3–6 replicates. Prism version 9.0 software (GraphPad Software, Inc., San Diego, CA, USA) was used to perform statistical analyses. One-way or two-way analysis of variance (ANOVA) followed by post-hoc multiple comparisons with Bonferroni correction were used for data comparisons when one or two influencing factors (e.g., different cell lines and/or different treatments) were involved for multi sample comparisons. Pearson correlation was used to determine linear relation between two parameters, the co-efficient of determination (*r*^2^) indicates the portion of the variation in one parameter that may be attributed to the other parameter. Dose-dependency between groups received treatments of different concentrations was tested using the nonparametric Jonckheere-Terpstra test following a significant one-way ANOVA and post-hoc multiple comparisons with Dunnett correction, as recommended [28]. A *p* value of less than 0.05 was considered statistically significant.

## 5. Conclusions

Here, we clearly identified TLR4, which is expressed on cancer cells, as a receptor for PAUF. We also demonstrated that PAUF’s tumor migration-promoting effects are exclusively through the TLR4/MyD88/NF-κB pathway. This study suggests the potential of the PAUF-TLR4 axis to become a therapeutic biomarker and target for the development of effective pancreatic cancer treatment.

## 6. Patents

The anti-PAUF antibody used in this study was registered as a patent in the following countries (regions): Korea (10-1856904, 2018), Taiwan (I703155, 2020), Australia (2017425111, 2020), Russia (2735102, 2020), South Africa (202001042, 2021), USA (11,046,779, 2021), Japan (7017581, 2022), Singapore (Patent No. not issued, 2022), Malaysia (Patent No. not issued, 2022). And the patent application is pending in another 15 countries (regions).

## Figures and Tables

**Figure 1 ijms-23-11414-f001:**
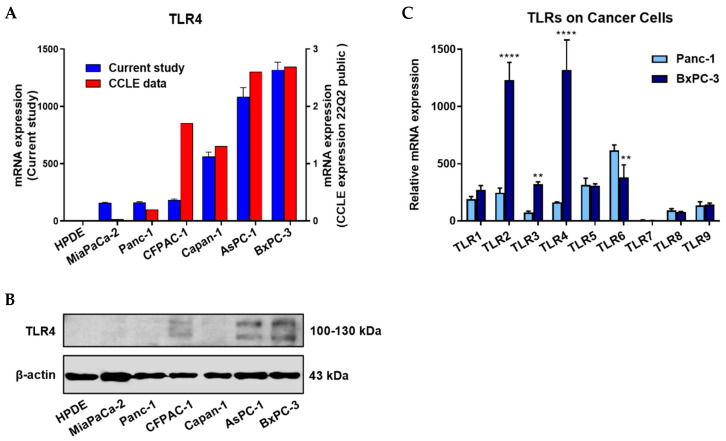
Positive TLR4 expression is detected in six different pancreatic cancer cell lines but not the HPDE normal pancreatic cell line. (**A**) mRNA expression levels of TLR4 quantified by RT-qPCR from the current study and retrieved from the Cancer Cell Line Encyclopedia (CCLE) Expression 22Q2 Public database. (**B**) Protein expression level of TLR4 assessed by Western blot analyses, the two bands of TLR4 appeared to be glycosylated (130 kDa) and deglycosulated (100 kDa) TLR4 [9,10]. (**C**) Relative mRNA levels of TLR1~9 in two representative pancreatic cancer cell lines. The mRNA expression levels were adjusted based on the two cell lines’ TLR4 data from (**A**). Data are represented as a mean ± SD from triplicates, ** *p* < 0.01, **** *p* < 0.0001 (indicating differences between Panc-1 and BxPC-3) were obtained from two-way ANOVA and post-hoc multiple comparisons with Bonferroni correction.

**Figure 2 ijms-23-11414-f002:**
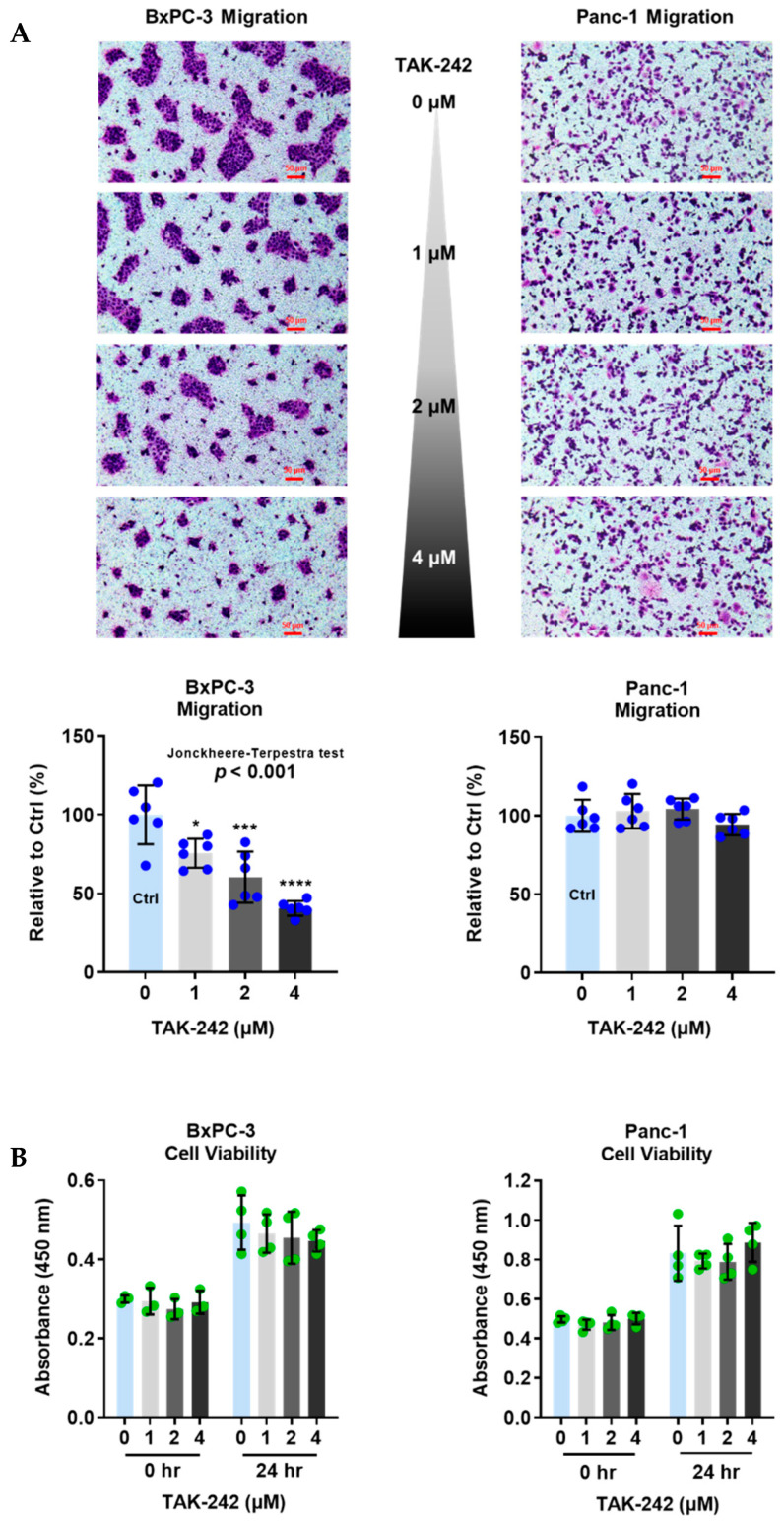
TLR4 inhibitor (TAK-242), at non-cytotoxic concentrations, reduces migration of pancreatic cancer cells expressing high level of TLR4 (BxPC-3). (**A**) Migration of BxPC-3 and Panc-1 cells, and (**B**) cell viability of BxPC-3 and Panc-1 cells measured after treated with TAK-242, a specific TLR4 inhibitor, at different concentrations for 24 h. Migration was presented as % of control, and cell viability was determined using WST-1 assay. Data are represented as mean ± SD from multiple replicates. The Jonckheere-Terpstra test was conducted to indicate the trend of cell migration’s change with increasing TAK-242 concentration after a significant multiple comparisons test (* *p* < 0.05, *** *p* < 0.001, **** *p* < 0.0001, compared to control, obtained from one-way ANOVA and post-hoc multiple comparisons with Dunnett correction). Scale bars, 50 μm.

**Figure 3 ijms-23-11414-f003:**
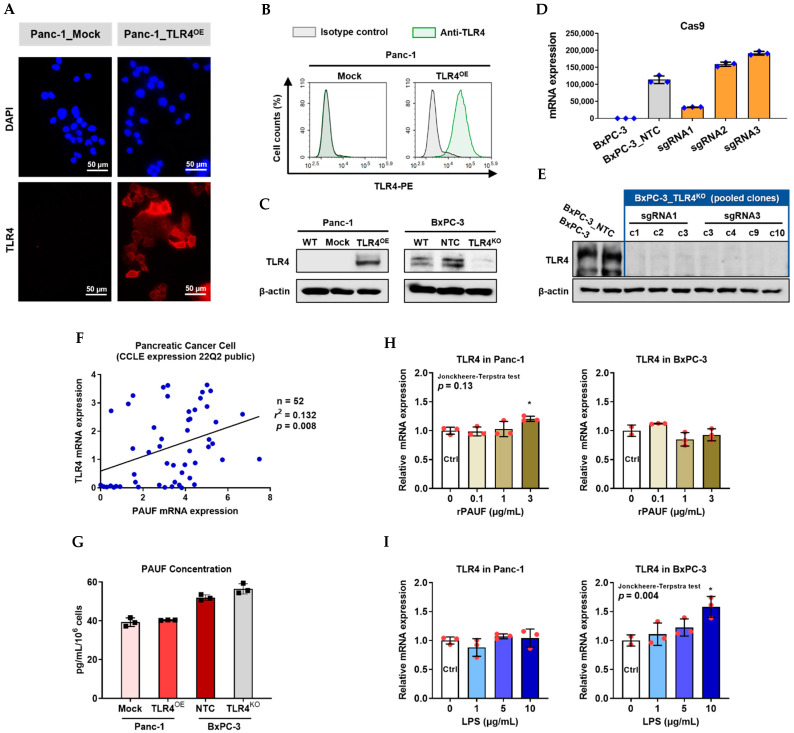
Successful generation of Panc-1 TLR4 overexpressed stable cell line (Panc-1_TLR4^OE^) and BxPC-3 TLR4 knockout stable cell line (BxPC-3_TLR4^KO^), and the impacts of TLR4 and PAUF expression on each other. Successful overexpression of TLR4 in Panc-1_TLR4^OE^ cell line was confirmed by (**A**) immunofluorescence, (**B**) flow cytometry, and (**C**) Western blot (SDS-PAGE gel: 10%). Successful knockout of TLR4 by CRISPR/Cas9 was confirmed in BxPC-3_TLR4^KO^ cells by (**D**) Cas9 mRNA expression and (**E**) Western blot (SDS-PAGE gel: 8%) against TLR4 in seven single clones with loss-of-function TLR4 mutations, which were pooled to form BxPC-3_TLR4^KO^ cells. And the knockout of TLR4 in the pooled cells was confirmed by Western blot and shown in (**C**). (**F**) The correlation of TLR4 and PAUF mRNA expression was analyzed using CCLE expression 22Q2 public data by Pearson correlation. (**G**) PAUF protein concentration in the four cell lines analyzed by sandwich ELISA. (**H**) Impacts of rPAUF (0, 0.1, 1, and 3 μg/mL) on TLR4 mRNA expression in Panc-1 and BxPC-3 cells. (**I**) Impacts of lipopolysaccharide (LPS, 0, 1, 5, and 10 μg/mL) on TLR4 mRNA expression in Panc-1 and BxPC-3 cells (LPS was used here as a positive control of PAUF). The dose-dependency of TLR4 mRNA expression on rPAUF/LPS concentration was tested by Jonckheere-Terpstra test, after a significant multiple comparisons test (* *p* < 0.05, compared to control, obtained from one-way ANOVA and post-hoc multiple comparisons with Dunnett correction). All data are presented as mean ± SD from triplicate independent experiments.

**Figure 4 ijms-23-11414-f004:**
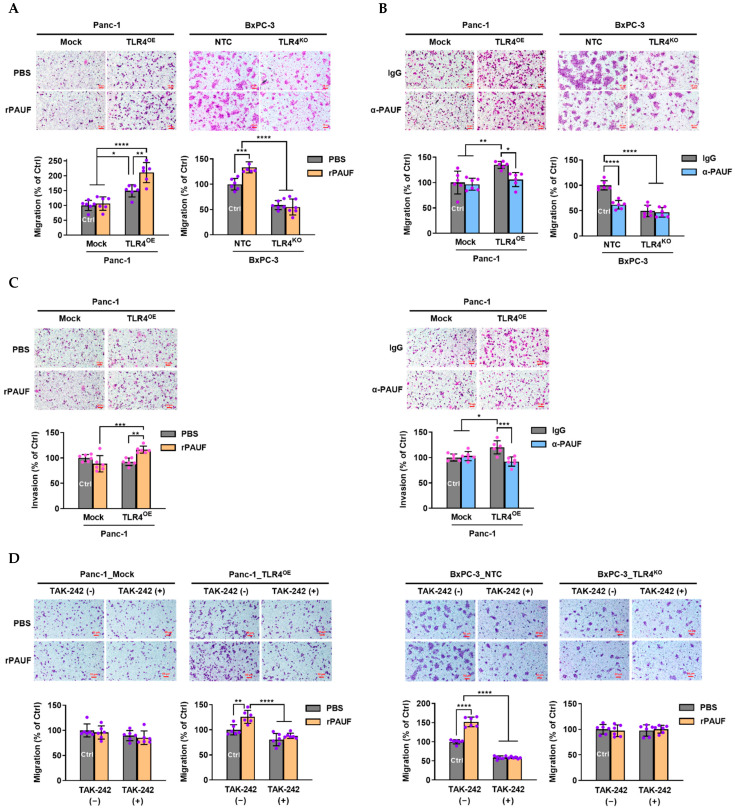
PAUF’s impacts on pancreatic cancer cell migration and invasion are exclusively dependent on TLR4. Migration of Panc-1_Mock and Panc-1_TLR4^OE^ were estimated using transwell assays after treatment of (**A**) recombinant PAUF (rPAUF, 0.5 μg/mL) or (**B**) PAUF antibody (α-PAUF, 20 μg/ml) for 24 h. (**C**) Invasion of Panc-1_Mock and Panc-1_TLR4^OE^ were determined using transwell assays after treatment of rPAUF (0.5 μg/mL) or α-PAUF (20 μg/ml) for 24 h. (**D**) Migration of pancreatic cancer cells induced by rPAUF in presence or absence of TAK-242, a TLR4 inhibitor. Data are represented as mean ± SD from multiple replicates. * *p* < 0.05, ** *p* < 0.01, *** *p* < 0.001, **** *p* < 0.0001 were obtained from two-way ANOVA and post-hoc multiple comparisons with Bonferroni correction. Scale bars, 50 μm.

**Figure 5 ijms-23-11414-f005:**
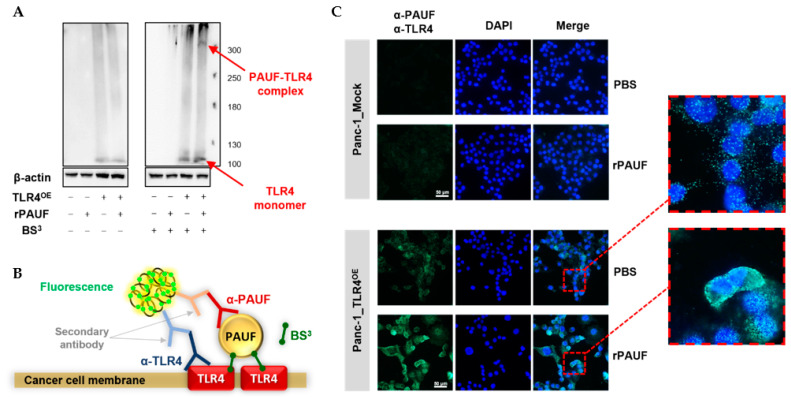
PAUF binds directly to TLR4 expressed on pancreatic cancer cells. (**A**) A PAUF-TLR4 complex was detected in Panc-1_TLR4^OE^ cells using Western blot after treatments of recombinant PAUF (rPAUF, 1 μg/mL) and crosslinker (BS^3^). (**B**) Schematic illustration of proximity ligation assay (PLA). (**C**) PLA was performed in Panc-1_Mock and Panc-1_TLR4^OE^ cells with or without treatments of rPAUF. Extended focus images for framed areas are shown. Scale bars, 50 μm.

**Figure 6 ijms-23-11414-f006:**
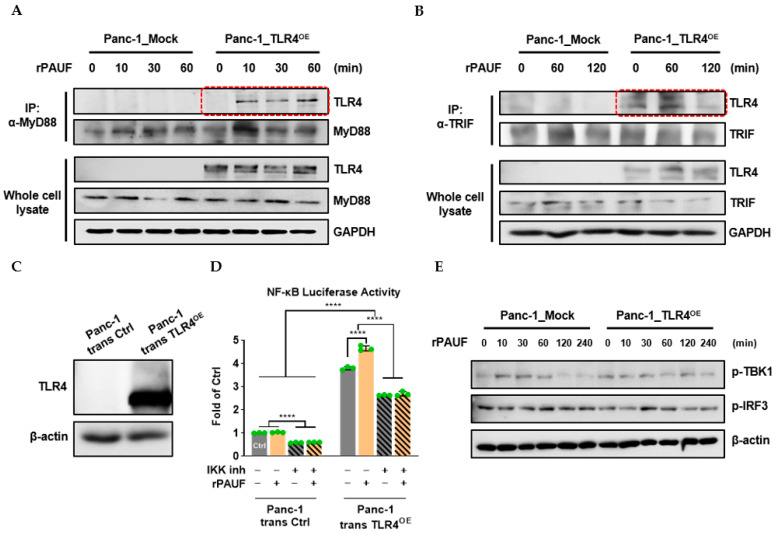
PAUF activates the MyD88-dependent, not the TRIF-dependent, TLR4 downstream pathway. (**A**) Panc-1_Mock and Panc-1_TLR4^OE^ cells were treated with rPAUF (0.1 μg/mL) for 0, 10, 30, and 60 min. Collected proteins were immunoprecipitated with MyD88 antibody (α-MyD88) using protein A agarose beads. Immunocomplexes were determined by Western blot against MyD88 and TLR4. (**B**) Panc-1_Mock and Panc-1_TLR4^OE^ cells were treated with rPAUF (0.1 μg/mL) for 0, 60, and 120 min. Collected proteins were immunoprecipitated with TRIF antibody (α-TRIF) using protein A agarose beads. Immunocomplexes were determined by Western blot against TRIF and TLR4. (**C**) Western blot showed that Panc-1_trans TLR4^OE^ cells were successfully engineered to transiently overexpress TLR4, compared to the control, Panc-1_trans Ctrl cells. (**D**) Panc-1_trans Ctrl and Panc-1_trans TLR4^OE^ cells were pretreated with or without IKK inhibitor XII (5 μM) for 4 h before rPAUF (0.1 μg/mL) or PBS treatment. After overnight culture, cells were collected for dual-luciferase activity assay. Data are represented as a mean ± SD from triplicates (**** *p* < 0.0001 were obtained from two-way ANOVA and post-hoc multiple comparisons with Bonferroni correction). (**E**) Panc-1_Mock and Panc-1_TLR4^OE^ cells were treated with rPAUF (0.1 μg/mL) for 0, 10, 30, 60, 120, and 240 min, and activation of TRIF-dependent TLR4 downstream molecules (p-TBK1 and p-IRF3) was estimated using Western blot.

**Table 1 ijms-23-11414-t001:** The primer sequences of the genes for mRNA expression quantification by qPCR.

Gene	Forward Primer (5′–3′)	Reverse Primer (5′–3′)
TLR1	GTTTTGTCTCCCAACTTTGTCC	TAGGAATGGAGTACTGCGGAAT
TLR2	CAAGCCCCTTTCTTCTTTAACAT	AGGAAGGTAAGTCCAGCAAAATC
TLR3	GCTTTAATCCCTTTGATTGCAC	AAAGGTAGTGGCTTGACAGCTC
TLR4	TTTCACCTGATGCTTCTTGCT	TCCTTACCCAGTCCTCATCCT
TLR5	TACAGCGAACCTCATCCACTTAT	ATTCTCTGAAGGGGTTTGATCTC
TLR6	ATCCTGCCATCCTATTGTGAGT	TTGCAGCTTCATAGCACTCAAT
TLR7	TTGGGGCTAGATGGTTTCC	TGAGGTTCGTGGTGTTCGT
TLR8	ATAGCAGGCGTAACACATCATCT	AATTCTACCAGGGACTTGCTTTC
TLR9	CCAAATCCCTCATATCCCTGT	ACAGTTGCCGTCCATGAATAG
GAPDH	CTTTGGTATCGTGGAAGGACTC	GTAGAGGCAGGGATGATGTTCT
Cas9	GGACTCCCGGATGAACACTA	TACCCTAAGCTGGAAAGCGA
PDL1	CACTACACAGCCCTCCTAA	GGAGACACTGTTTCTTCAGC

**Table 2 ijms-23-11414-t002:** The oligonucleotide sequences of the TLR4 sgRNAs and scrambled sgRNA.

sgRNA	Forward Oligo (5′–3′)	Reverse Oligo (5′–3′)
sgRNA1	CACCGCCTGCGTGAGACCAGAAAGC	AAACGCTTTCTGGTCTCACGCAGG
sgRNA2	CACCGGCGCGAGGCAGACATCATCC	AAACGGATGATGTCTGCCTCGCGC
sgRNA3	CACCGTAGCTGCCTAAATGCCTCAG	AAACCTGAGGCATTTAGGCAGCTA
Scrambled sgRNA	CACCGTTCCGCGTTACATAACTTA	AAACTAAGTTATGTAACGCGGAAC

## Data Availability

Data available on request from the authors.

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
