# Peer review of "PAUF Induces Migration of Human Pancreatic Cancer Cells Exclusively via the TLR4/MyD88/NF-κB Signaling Pathway"

_ijms, 2022, doi:10.3390/ijms231911414_

Round 1
Reviewer 1 Report
The authors provide new information on the role of PAUF in pancreatic cancer cell migration and its molecular regulation by the TLR4/MyD88/NF-kB signaling pathway. Overall, this article is clear and rather easy to understand, the materials and methods are well described. However there are some minor comments:
- In figure 1A and 1B, the expression of TLR4 protein in the western blot does not appear to match the mRNA expression for the Capan-1 cell line. In addition, there are two bands on the presented western blot, and others non-specific bands on the original blot, suggesting a lack of sensitivity of the antibody used.
- In Figure 2, do the authors have the transwell assay images associated with the migration data? (To put them on the article or in a supplementary figure).
- In 2.3 part, why did the authors decide to perform 3 different methods to validate the TLR4 protein expression of their model when they do not perform an mRNA level study?
- In figure 3C and 3E, these two western blot target the same protein with the same antibody used, but the images shown are very different.
- Figure 3G is not cited in the article, perhaps it would be on line 349?
- Figure 5 and 6, why did the authors decide to perform these different experiments only in the Panc-1 model? A small explanation would be judicious to bring
- Lines 290-294, do the authors have any idea how to explain the difference in TLR4 expression levels in the different pancreatic cancer cell lines? It might be interesting to add some minor information about this.
- Line 330: the study of invasion effects might be interesting to add on the article.
- Line 359: Do the authors have any protein data to support their hypothesis?
- Line 363: The authors’ explanation for the observed non-specific signal is limited. Indeed, there is no signal detected in Panc-1-Mock cells but there is in Panc-1-TLR4OE cells. Do the authors have another hypothesis? Is it possible that Panc-1-TLR4OE may have and endogenous expression of PAUF?
Author Response
Reviewer 1
The authors provide new information on the role of PAUF in pancreatic cancer cell migration and its molecular regulation by the TLR4/MyD88/NF-kB signaling pathway. Overall, this article is clear and rather easy to understand, the materials and methods are well described. However there are some minor comments:
Dear reviewer, thank you very much for your time and efforts in reviewing our manuscript. We have looked through all the issues you pointed out, we tried to dress your comments point by point and made revision accordingly. Your review has helped us improve the article greatly, and we hope it may be suitable for publication now.
- In figure 1A and 1B, the expression of TLR4 protein in the western blot does not appear to match the mRNA expression for the Capan-1 cell line.
Re: Thank you for your careful review. The lack of correlation between protein and mRNA expression in cancer cells/samples was in fact frequently observed, for example: In prostate cancer, the lack of correlation was reported for MMP2, MMP9, TIMP-1 (Ref. 1). In lung cancer, the lack of correlation was observed for multiple (n=98) genes (Ref. 2).
In addition, in this study, the TLR4 protein and mRNA expression was generally well correlated and in agreement with CCLE data, and the selection of two representative cell lines with low and high TLR4 expression (Panc-1 and BxPC-3) for further study was therefore justified.
In addition, there are two bands on the presented western blot, and others non-specific bands on the original blot, suggesting a lack of sensitivity of the antibody used.
Re: In this study, both the monoclonal (santa cruz, #sc-13593) and polyclonal (R&D, #AF-1478) anti-TLR4 antibodies were tested. In weighing up the “pros” and “cons” of the two antibodies, we decided to choose the polyclonal antibody which showed much more desired sensitivity for Western blot analyses, although it showed less favorable specificity. However, although multiple non-specific bands were observed following the use of polyclonal antibody, they did not interfere the detection of the TLR4 bands.
As you mentioned, two TLR4 bands were observed at approximately 100 kDa and 130 kDa, which are estimated to be glycosylated and deglycosylated TLR4 bands, respectively (Ref. 3 and 4). We added these references for Figure 1B.
Ref. 4
- In Figure 2, do the authors have the transwell assay images associated with the migration data? (To put them on the article or in a supplementary figure).
Re: Thank you for your good advice, we have added the migration images as a supplementary figure (also attached as below) and revised the result accordingly.
Figure 2A
- In 2.3 part, why did the authors decide to perform 3 different methods to validate the TLR4 protein expression of their model when they do not perform an mRNA level study?
Re: Thank you for your question. Since an increase in TLR4 protein level of the Panc-1_TLROE was clearly indicated using Western blot analyses, we did not measure the mRNA expression level. In addition, this study aimed to find out whether PAUF can bind to TLR4 expressed on the cancer cell membrane, however, measuring mRNA expression level is unable to tell whether the overexpressed TLR4 was due to increased TLR4 expression on cell membrane or not. By contrast, IHC and Flow Cytometry experiments clearly showed that TLR4 was overexpressed on the cancer cell membrane of Panc-1_TLR4OE cells.
- In figure 3C and 3E, these two western blot target the same protein with the same antibody used, but the images shown are very different.
Re: Thank you for your careful review. The reason for the two bands of TLR4 in 3E being more far part than in 3C is most likely because the percentage of SDS-PAGE gel are different, which is 10% in 3C, and 8% in 3E. Despite the different looks of the bands, the bands size appeared to be the same (100 and 130 kDa)
To avoid the potential confusion, we added the gel info into the legend of Figure 3.
- Figure 3G is not cited in the article, perhaps it would be on line 349?
Re: Thank you for pointing out this issue for us. We have adjusted the sequence of Figure 3G to 3F and added the description of Figure 3F into the Results (part 2.3) as follows (line 132):
In the PC cell lines from the CCLE database (n = 52), there is a weak positive correlation (r = 0.363, p = 0.008) between TLR4 and PAUF mRNA expression (Figure 3F).
- Figure 5 and 6, why did the authors decide to perform these different experiments only in the Panc-1 model? A small explanation would be judicious to bring
Re: Thank you for your question. For study simplicity, we used only Panc-1 cell. The rationale supporting this simplification is that both BxPC-3 and Panc-1_TLR4OE cell lines express high level of TLR4 (as shown in Figure 3C, attached below), and the only difference between the Panc-1_TLR4OE/_Mock pair, and between the BxPC-3_NTC/_TLR4KO pair is the TLR4 expression level. Therefore, to study if PAUF activated TLR4 signaling pathway, either the Panc-1 or the BxPC-3 pair is considered suitable. Moreover, because we already know that TLR4 is overexpressed on the cell surface of Panc-1_TLR4OE cells (Figure 3A and 3B), we selected this cell line pair over the BxPC-3 pair.
|
|
|
Figure 3
- Lines 290-294, do the authors have any idea how to explain the difference in TLR4 expression levels in the different pancreatic cancer cell lines? It might be interesting to add some minor information about this.
Re: Thank you for your comment and good advice. We do not know the exact mechanism of the diversity in TLR4 expression among different cancer cells. But according to the well-cited review article published in 2017 by Li et al. (Ref. X, which was also cited in the manuscript), TLR4 plays multiple important roles in shaping the tumor microenvironment (TME) via interacting with dendritic cells, T cells, MDSCs, endothelial cells etc. Given the complexity of TME, it is expected that TLR4 expression can vary vastly between individuals as well as cancer types.
Following your advice, we added more discussion as follows (line 317)
The vastly varied TLR4 expression was expected as TLR4 plays multiple roles in shaping the tumor microenvironment [12], and the latter is famously known of its complexity.
- Line 330: the study of invasion effects might be interesting to add on the article.
|
|
|
|
Re: Thank you for your good advice. We have added the invasion images, as well as other migration images, into the Figure 4 as below:
- Line 359: Do the authors have any protein data to support their hypothesis?
Re: Thank you for your comment. The lack of correlation between protein and mRNA expression in cancer cells/samples was in fact frequently observed, for example: In prostate cancer (17 samples), the lack of correlation was reported for MMP2, MMP9, TIMP-1 (Ref. 1). In lung cancer (196 samples), the lack of correlation was observed for multiple (n=98) genes (Ref. 2). We have added these references into the main text.
Another possible explanation is that LPS and PAUF activate different pathways and therefore they showed different patterns in impacting TLR4 mRNA expression. We have added additional discussion as follows:
We did not observe upregulation of TLR4 by rPAUF in BxPC-3 cells though, probably because mRNA expression results do not always correlate well with protein expression [14,15].
- Line 363: The authors’ explanation for the observed non-specific signal is limited. Indeed, there is no signal detected in Panc-1-Mock cells but there is in Panc-1-TLR4OE cells. Do the authors have another hypothesis? Is it possible that Panc-1-TLR4OE may have and endogenous expression of PAUF?
Re: Thank you for your insightful comment. It is unlikely that the non-specific signals observed in PBS-treated Panc-1_TLR4OE cells not in the PBS-treated Mock cells was caused by endogenous expression of PAUF, because:
1) The overexpression of TLR4 does not impact the PAUF concentration as shown in Figure 3G, and
2) in this experiment, we removed all medium from cells before treatment of PBS (or rPAUF). After that, cells were incubated with anti-PAUF and anti-TLR4 antibodies at 4℃. It is unlikely any endogenous PAUF remained in the cell culture medium or was released from the cell.
Figure 3G
References:
- Lichtinghage R.; Musholt P.B.; Lein M. Different mRNA and protein expression of matrix metalloproteinases 2 and 9 and tissue inhibitor of metalloproteinases 1 in benign and malignant prostate tissue. Eur Urol. 2002, doi: 10.1016/s0302-2838(02)00324-x.
- Chen G.; Gharib T.G.; Huang C.C. Discordant protein and mRNA expression in lung adenocarcinomas. Mol Cell Proteomics. 2002, doi: 10.1074/mcp.m200008-mcp200.
- Correia J.S.; Ulevitch R.J. MD-2 and TLR4 N-linked glycosylations are important for a functional lipopolysaccharide receptor. J Biol Chem. 2002, doi: 10.1074/jbc.M109910200. Epub 2001 Nov 12.
- Ohnishi T.; Muroi M.; Tanamoto K.I. MD-2 is necessary for the toll-like receptor 4 protein to undergo glycosylation essential for its translocation to the cell surface. Clin Diagn Lab Immuno. 2003, doi: 10.1128/cdli.10.3.405-410.2003.

Reviewer 2 Report
Dr. Youn and the colleagues demonstrated that pancreatic adenocarcinoma up-regulated factor (PAUF) promotes migration of human pancreatic ductal adenocarcinoma cell lines through TLR4/MyD88/NF-κB pathway. The detailed analyses supports the phenomena, but major revisions are required for publication.
- PD-L1 data were not associated with the title and conclusion completely. Please delete Fig. 7 in the manuscript and the figures should be moved to Supplementary data.
- The authors cited such as (Figure 1A, B) in the section of Discussion. These are so strange and look like repeating results. Also, the authors should describe their short summary in the final sentence of each paragraph.
- Graphic Abstract: Schema in the absent status of PAUF should be addressed. Or, the two pathways TRIF and MyD88 should be summarized as the presence of PAUF.
Minor concerns:
- Introduction (p2, line 63): LPS and PD-L1 should be spelled out.
- Fig. 3H and I: Strange statistical analysis in the histogram. Is there a significant difference between groups?
- Fig.5C: Why can the authors conclude that the signals of fluorescence are localized on plasma membrane, but not cytoplasm.
Author Response
Reviewer 2
Dr. Youn and the colleagues demonstrated that pancreatic adenocarcinoma up-regulated factor (PAUF) promotes migration of human pancreatic ductal adenocarcinoma cell lines through TLR4/MyD88/NF-κB pathway. The detailed analyses supports the phenomena, but major revisions are required for publication.
Dear reviewer, thank you very much for your time and efforts in reviewing our manuscript. We have looked through all the issues you pointed out, we tried to dress your comments point by point and made revision accordingly. Your review has helped us improve the article greatly, and we hope it may be suitable for publication now.
- PD-L1 data were not associated with the title and conclusion completely. Please delete Fig. 7 in the manuscript and the figures should be moved to Supplementary data.
Re: Thank you for your good advice. We have moved the figure to supplementary data. In addition, we deleted the PD-L1 contents from the Introduction part.
- The authors cited such as (Figure 1A, B) in the section of Discussion. These are so strange and look like repeating results.
Re: Thank you for your comments. We hoped citing the figures may help our readers quickly spot the contents for discussion. However, we may have cited too many figures, therefore in the revision, we reduced the citations.
Also, the authors should describe their short summary in the final sentence of each paragraph.
Re: Thank you for your good advice. We have added one sentence to outline what we did and what the results are in the beginning of each part of the results (except 2.2 which we had the summary already).
2.1. TLR4 was expressed in PC cells but not in normal pancreatic cells
Using RT-qPCR and Western blot, we showed that TLR4 is expressed in all the PC cell lines, at various levels, but not in the normal pancreatic cell line, and the expression of other TLR subtypes also differed significantly between different PC cell lines.
2.2. Endogenous ligands induced PC cell migration via TLR4
Using TAK-242, a chemical TLR4 inhibitor, we showed that PC cell-secreted ligands induced PC migration via TLR4 by transwell assay.
2.3. Successful generation of TLR4 overexpressed and knockout PC cells
At mRNA and protein expression levels, we confirmed TLR4 overexpression and knockout in four engineered stable PC cell lines. And we found that TLR4 expression does not impact PAUF concentration, but PAUF concentration tends to upregulate TLR4.
2.4. PAUF induced PC cell migration was dependent on TLR4 expression
Using cell migration/invasion assays, we observed that PAUF was able to induce PC cell migration and invasion, but only in cells with high expression of normal function TLR4.
2.5. PAUF bound to TLR4 on the surface of pancreatic cancer cells
Using Western blot and immunofluorescence assay, we observed direct binding of PAUF and TLR4 on the PC cell surface.
2.6. PAUF activated TLR4 through MyD88-dependent signaling pathway.
Using immunoprecipitation, Western blot, and reporter gene assays, we discovered that PAUF activated MyD88 but not TRIF of TLR4 downstream signaling pathway.
2.7. PAUF up-regulated programmed death-ligand 1 (PD-L1) expression in cancer cell cytoplasm.
After we confirmed that PAUF selectively simulated TLR4/MyD88 signaling pathway, we further investigated if PAUF can up-regulate PD-L1 expression, because a recent study showed that LPS can mediate PD-L1 upregulation in PDAC via the same pathway.
Using flow cytometry analyses and RT-qPCR, we discovered that similar to LPS, PAUF caused an increase in overall PD-L1 expression in PC cells, but neither LPS nor PAUF increased PD-L1 expression on PC cell surface.
- Graphic Abstract: Schema in the absent status of PAUF should be addressed. Or, the two pathways TRIF and MyD88 should be summarized as the presence of PAUF.
Re: Thank you for your good advice. We agree it is informative to show a status of no PAUF stimulation, therefore we have revised the Graphic Abstract as follow:
Minor concerns:
- Introduction (p2, line 63): LPS and PD-L1 should be spelled out.
Re: Thank you for your careful review, we have added full term for LPS (Lipopolysaccharide) and PD-L1 (Programmed death-ligand 1) into the main text at their first appearances.
- Fig. 3H and I: Strange statistical analysis in the histogram. Is there a significant difference between groups?
Re: Thank you for your comment. We conducted Pearson correlation analyses for these data because rather than differences between groups, the Pearson correlation can show if there is any does-dependent effect of PAUF on TLR4 mRNA expression. To avoid any confusion, we have revised the figure legend.
The dose-dependency between rPAUF/LPS and TLR4 mRNA expression was tested by Pearson correlation, p < 0.05 indicates significant dose-dependency, and r2 is the coefficient of determination, indicating the percentage of variation in mRNA expression can be explained by rPAUF or LPS treatment.
- Fig.5C: Why can the authors conclude that the signals of fluorescence are localized on plasma membrane, but not cytoplasm.
Re: Thank you for your insightful comment. The signals observed in PBS-treated Panc-1_TLR4OE cells not in the PBS-treated _Mock cells is unlikely localized in cytoplasm because:
1) the signal emission in the PLA assay requires linkage of two molecules using a chemical crosslinker BS3, which is a membrane-impermeable agent, and
2) no permeabilization step was conducted in this experiment, therefore antibodies unlikely entered the cytoplasm.

Round 2
Reviewer 2 Report
The revised manuscript is improved according to the reviewer’s comments, but the statistical analyses especially Fig.2A, Fig.3H and 3I must be double checked by statistician because Pearson correlation test is generally used for scatter plots like Fig.3F in the revised version.
Author Response
Re: Dear reviewer, thank you very much for second round of review. Based on your comment, we have revised the statistical method used for data analyses in the three figures and the Statistics section as well (Revised parts have been highlighted).
To test for not only differences between groups but also for dose-dependency, we have then used a method recommended by the reference shown below, where it says:
“Therefore, we suggest to examine the data for the difference between each dose group and control by Dunnett’s test and then examine the data by Jonckheere’s trend test for dose related relationship.”
Fig.2A Fig.3H Fig. 3I
Reference:
Katsumi Kobayashi, et al. Determination of dose dependence in repeated dose toxicity studies when mid-dose alone is insignificant. J Toxicol Sci. 2012;37(2):255-60. doi: 10.2131/jts.37.255.
